# Descriptive and Functional Genomics in Acute Myeloid Leukemia (AML): Paving the Road for a Cure

**DOI:** 10.3390/cancers13040748

**Published:** 2021-02-11

**Authors:** Hélène Pasquer, Maëlys Tostain, Nina Kaci, Blandine Roux, Lina Benajiba

**Affiliations:** 1Université de Paris, APHP, Hôpital Saint-Louis, 75010 Paris, France; helene.pasquer@inserm.fr (H.P.); maelys.tostain@etu.u-paris.fr (M.T.); nina.kaci@inserm.fr (N.K.); blandine.roux@inserm.fr (B.R.); 2INSERM UMR 944, Institut de Recherche Saint-Louis, 75010 Paris, France

**Keywords:** acute myeloid leukemia, genomics, molecular landscape, prognosis, physio-pathological models, targeted therapies

## Abstract

**Simple Summary:**

Acute myeloid leukemia (AML) accounts for 7.6% of hematopoietic malignancies with a long-term survival of less than 20%. Better understanding its physiopathology and finding new treatments remain important issues. In the current review, the authors discuss how genetic engineering technologies improvement allowed a better genetic characterization of AML. Such molecular dissection of the AML genome had two direct clinical impacts: a prognostic contribution by defining a new molecular classification of AML which guides therapeutic regimen intensity, and a therapeutic impact by allowing the identification of new therapeutic targets. New genome editing tools and animal models have also paved the way for a better understanding of AML leukemogenesis. Their impact is also summarized in this review. The combination of descriptive and functional genetics may ultimately be the key to improving the prognosis of this dismal disease.

**Abstract:**

Over the past decades, genetic advances have allowed a more precise molecular characterization of AML with the identification of novel oncogenes and tumor suppressors as part of a comprehensive AML molecular landscape. Recent advances in genetic sequencing tools also enabled a better understanding of AML leukemogenesis from the preleukemic state to posttherapy relapse. These advances resulted in direct clinical implications with the definition of molecular prognosis classifications, the development of treatment recommendations based on minimal residual disease (MRD) measurement and the discovery of novel targeted therapies, ultimately improving AML patients’ overall survival. The more recent development of functional genomic studies, pushed by novel molecular biology technologies (short hairpin RNA (shRNA) and CRISPR-Cas9) and bioinformatics tools design on one hand, along with the engineering of humanized physiologically relevant animal models on the other hand, have opened a new genomics era resulting in a greater knowledge of AML physiopathology. Combining descriptive and functional genomics will undoubtedly open the road for an AML cure within the next decades.

## 1. Introduction

Acute myeloid leukemia (AML) is a clonal myeloid neoplasm in which abnormal proliferation and impaired differentiation of stem and progenitor cells in the bone marrow (BM); impede normal haematopoiesis and results in the accumulation of immature blasts. Although some patients develop secondary AML from other haematopoietic disorders or AML-related therapy, the majority of them present de novo AML [1]. Despite the high rate of complete remission after initial chemotherapeutic treatment, the relapse rate is still significant. This disease, therefore, remains devastating with about 18,000 new cases every year in Europe and a five-year overall survival of only 17% [2]. AML classification was historically based on cell morphology with eight nosological subgroups (M0 to M7) [3]. Over the past decades, this classification has been repeatedly revised accompanying the advances of our AML pathogenesis understanding [4,5,6,7,8].

In this review, we examine how recent advances in genomics, hematopoietic stem cells biology and physiopathologically relevant in vivo modelling have resulted in a better molecular characterization and leukemogenesis understanding. We describe the clinical impact of this newly acquired knowledge, including, on one hand, identification of novel genetic markers allowing more accurate patient prognostic stratification and treatment decisions, and, on the other hand, opening the road for a new therapeutic era in AML.

## 2. The AML Genome: Descriptive Genomics

### 2.1. Genomic Technological Advances and Their Contribution to AML Molecular Characterization

AML like every cancer, is defined by an out-of-control cellular growth due to genetic alterations accumulation in the cell’s genome. Throughout the last decades, genetic engineering technology improvement has allowed a more precise molecular dissection of the AML genome and the discovery of new genetic abnormalities.

The first AML genetic abnormalities were identified using standard cytogenetics. Through analysis of the structure and number of chromosomes, researchers were able to highlight chromosomal rearrangements possibly involved in the leukemic process. In 1976, Oshimura and his team observed a translocation between the long arms of chromosomes 8 and 21 (t (8;21) (q22;q22)) in several AML patients [9]. The relevance of this genomic alteration in leukemogenesis and its association with AML were validated in 1983 by Larson et al. who described the same rearrangement in similar AML cases, mostly of the M2 morphological subtype. The same group also discovered another common cytogenetic alteration in AML patients characterized by an inversion in chromosome 16 (inv(16)(p13;q22)), leading to the emergence of the CBFb-Myh11 fusion gene [10].

With the invention of the Sanger sequencing method by Frederick Sanger (1980 Nobel prize), enabling entire cell genome sequencing nucleotide by nucleotide, AML genomics entered a new era. Comparative studies of normal hematopoietic cells and leukemic cell DNA resulted in the identification of new mutations in AML patients with normal karyotypes. The Sanger sequencing method results in the replacement of a normal deoxyribonucleotide during DNA replication by a marked dideoxyribonucleotide, which causes the process’s arrest. Random incorporation of either the deoxyribonucleotide or the dideoxyribonucleotide by the DNA polymerase ultimately allows the entire genome’s reconstitution. Using this technique, Nakao and colleagues were able to discover an internal tandem duplication in the *FLT3* gene of the AML cell genome in 1996 [11]. Between 1995 and 1999, mutations in several other genes such as *TP53*, *N/K-RAS*, *MLL*, *WT1* and *KIT* were associated with AML [12,13,14,15]. Later on, a mutation in the *CEBPA* gene was reported by Thomas Pabst et al. [16]. In 2005, mutations in the *NPM1* gene, responsible for NPM1 protein mislocation from the nucleus to the cytoplasm, were described by Falini et al. [17].

In 1992, Dan Pinkel invented a new genetic technology, called CGH (Comparative Genomic Hybridization) array. CGH allows the identification of genomic copy number aberrations such as duplications or deletions, by comparing two fluorescently labelled genomic DNA portions’ hybridization. This method is more precise than standard karyotyping and can be more practical although it is less exhaustive than Sanger sequencing. Based on a similar approach, SNP (Single Nucleotide Polymorphisms) arrays can detect variations of a single nucleotide in a defined gene. These two techniques were used in 2009 by Delhommeau and colleagues and in 2010 by Ernst et al., leading to the discovery of mutations in the TET2 and EZH2 genes, respectively, in myeloid diseases [18,19].

In the early 2000s, Next Generation Sequencing (NGS), or High-Throughput Sequencing (HTS) were developed, opening the road for a prosperous discovery era in cancer genomics. Compared to Sanger sequencing, NGS allows faster genome sequencing through the use of custom-made sequencing libraries. Using this technique, the AML’s genome of AML was completely sequenced for the first time in 2008 by Ley et al. [20]. HTS, therefore, led to the discovery of new recurring mutations in AML. In 2010, Mardis et al. found mutations in the *IDH1, IDH2, DNMT3A* and *UTX* genes [21], while mutations in *GATA2, PFH6* and *BCOR* genes were identified later in 2011 [22,23,24].

More recently, a new generation of HTS allowed genome sequencing at a single cell level [25]. In 2019, Van Galen et al. and Potter et al. used this powerful technology to decipher the clonal heterogeneity present in one AML patient. They attempted to describe the architecture and phylogenetic order of mutations emergence to better characterize AML’s clonal evolution [26,27].

Figure 1 summarizes the parallel discovery roads of AML’s molecular landscape dissection and genomic technological advances.

### 2.2. The Main AML Recurrently Mutated Genes: Description and Function

Genomic technological development unraveled a heterogeneous mutational landscape and opened the road for a more precise understanding of leukemogenic processes.

The transformation of normal hematopoietic cells to AML has been based on the double hit model for a long time. This physiopathological model asserts that the occurrence of AML is the consequence of the association of two genetic events: class I mutations, which confer independent proliferation properties to AML cells, and class II mutations responsible for hematopoietic differentiation blockade. Gilliland and colleagues described this model for the first time in 2002 [28]. They revealed a cooperation between the PML/RARa fusion gene and the FLT3-ITD mutation in the induction of an Acute Promyelocytic Leukaemia (APL)-like disease in a murine model: PML/RARa expression causes a differentiation blockade, whereas activating mutations in the *FLT3* gene are responsible for cytokines independent growth of hematopoietic cells. The occurrence of only one of these two genetic events induces a myeloproliferative syndrome but is not sufficient to induce overt leukemia. For the first time, these observations highlighted the need of two types of genetic abnormalities for AML genesis. The first event, defined as a class I mutation, induces an excess of proliferation while the second one, named a class II mutation, blocks myeloid differentiation. This theory was subsequently validated on other AML subtypes. For example, in core binding factors (CBF) AML, the disruption of *CBFa/b* genes confers a differentiation blockade. This founding genetic event is often associated with additional mutations harbored by signaling proteins coding genes responsible for increased proliferation such as *FLT3* or *KIT* [29].

Recent whole-genome sequencing studies revealed AML genomic complexity, thereby out-fashioning the canonical AML two-hit model. Indeed, Grove et al. [30] showed that each AML genome harbors approximately 17 mutations. Among these mutations, we distinguish founding drivers and secondary passenger events. Interestingly, the genome of AML is less mutated than that of solid tumors (~100 mutations per genome). This peculiarity may explain why AML is thought to be less immunogenic than other malignancies [31].

In a large study among 1540 AML patients, Papaemmanuil et al. [32] identified more than 5000 driver mutations in 76 genes (of which the most frequent are *FLT3*-ITD (30%), *NPM1* (29%) and *DNMT3A* mutations (23%)). Among these mutations, some are commonly associated such as co-occurrence of *IDH1* or *IDH2* mutations with *NPM1* mutations and co-occurrence of *DMT3A* mutations with *NPM1*, *FLT3* and *IDH* mutations. Other mutations, such as *IDH1, IDH2* or *TET2* mutations, are rather mutually exclusive.

Grove et al. [30] classified the recurrent AML mutations into nine functional categories. This classification is based on the functional role of each gene. The first category includes fusion genes, which induce the differentiation blockade, such as PML-RARa or MYH11-CBFb and RUNX1-RUNX1T1 found in the APL or CBF AML subtypes, respectively. Categories 2 and 3 contain myeloid transcription factors and tumor suppressor genes respectively. Genes involved in RNA splicing form the 4th category while category 5 includes DNA modifier genes (DNA methylation, etc.). The nucleolar phosphoprotein B23 (*NPM1*) gene, coding for a nucleolar chaperon protein involved in ribosomal synthesis and cellular proliferation, is the sole gene of the 6th category. Chromatin modifiers are grouped in category 7. Category 8 is composed of cohesin genes, encoding a protein complex that maintains the two chromatids together during mitosis. Signal transduction genes implicated in cellular proliferation form category 9. The cellular function of each category is summarized in Figure 2, where examples of frequently AML mutated genes are also highlighted for each functional family.

Fusion genes are responsible for a blockage of differentiation. Mutations of in myeloid transcription factors are implied in cell proliferation, whereas genetic alterations of tumor suppressor genes induce blockage of apoptosis. Abnormalities in spliceosome genes cause synthesis of new pre-mRNAs, implied in leukemogenesis. Mutations in *NPM1* gene are responsible for the synthesis of an abnormal nucleolar chaperon protein, which causes dysfunctions on ribosomal synthesis and cellular proliferation. Genetic alterations in cohesin genes lead to asymmetric division of chromosomes during mitosis. Mutations of chromatin modifiers or DNA modification genes are responsible for abnormal up and down regulation of genes implied in AML leukemogenesis.

### 2.3. Leukemogenesis and AML Clonal Evolution: From Preleukemic State to Relapse

Advances in genetics led to a better understanding of AML leukemogenesis and clonal evolution, from the preleukemic state to relapse.

In AML, around 20% of cases evolve from a pre-existing hematological malignancy such as Myelodysplastic Syndromes (MDS) or Myeloproliferative Neoplasms (MPN) [1]. Founder mutations present in preleukemic cells of MDS or MPN often remain present in AML blasts. However, most of AML cases arise de novo without any prior clinical perturbation. Recent studies revealed that de novo AML could develop from a preleukemic state found in clinically healthy individuals. This preleukemic state, named CHIP (Clonal Haematopoiesis of Indeterminate Potential) or ARCH (Age-Related Clonal Haematopoiesis) is a common phenomenon linked to aging. Aged hematopoietic stem cells (HSCs) can, therefore, acquire a unique somatic mutation and form a clonal subpopulation derived from a single founding cell.

Studies in the 90s measuring X-inactivation ratios identified a nonrandom X-inactivation pattern in blood cells, particularly within the myeloid compartment [33,34]. This biased phenotype was the initial clue to clonal evolution in a subset of hematopoietic cells derived from the same founding clone. The molecular hypothesis behind this inactivation bias was the acquisition of somatic mutations conferring a growth advantage [35]. 

This hypothesis was confirmed by Busque et al. in 2013 [36], who revealed, using whole genome sequencing, the recurrent occurrence of somatic *TET2* mutations in healthy elderly individuals. This concept of CHIP has been reported in many other studies since [37,38,39].

Clonal haematopoiesis with somatic mutations increases with age. It concerns approximately 10% of healthy individuals older than 65 years. There are many mutations involved in clonal haematopoiesis. The most frequent affect three epigenetic regulators (*DNMT3A, ASXL1*, and *TET2*). Mutations in signaling proteins (*JAK2*), spliceosome components (*SF3B1* and *SRSF2*), or members of the DNA damage response pathway (*TP53* and *PPM1D*), are also found in clonal haematopoiesis.

This latter process increases the risk of hematological malignancies development (hazard ratio: 11 to 13) and is also associated with coronary heart disease (Hazard Ratio: 2.0) [37,38,39].

Recently developed single cell technologies have also greatly improved the characterization of genetic heterogeneity in overt AML. As described earlier, AML is a genetically complex and heterogeneous disease with each patient harboring an extremely rare combination of somatic mutations, as illustrated by Döhner et al. [8]. Genetic heterogeneity within a single patient represents another layer of complexity. Indeed, AML is an oligoclonal disease with multiple genetically distinct subclones coexisting with the dominant clone at time of diagnosis [27,40,41]. In addition to this genetic heterogeneity, a functional heterogeneity has also been described. Klco et al. showed that AML patient xenografts samples were predominantly composed of a single genetically-defined subclone; suggesting variable subclonal engraftment potential in immune deficient mice [42].

New genetic tools have also led to a further comprehension of relapse mechanisms and clonal evolution in AML. Kronke et al. [43], reported an increased genomic complexity at time of relapse compared to diagnosis, and Ding et al. proposed a dual mechanistic explanation for AML relapse. The first hypothesis relies on the survival advantage of the founding clone resulting from acquisition of new mutations, while the second one states that a subclone survives initial therapy and gains additional mutations allowing its clonal expansion [44].

All these steps of AML leukemogenesis are summarized in Figure 3.

Schematic representation of the different stages of AML. Each star represents a new acquired somatic mutation. The first step is the occurrence of an acquired somatic mutation in an HSC leading to a clonal proliferation, a pr-leukemic state named CHIP. This preleukemic state can either never evolve or lead to an AML with second acquired mutation. During AML evolution, initial cells acquire additional mutations leading to a subclonal disease. After AML treatment, in some cases relapse is observed. It occurs from a founding cell or a subclonal cell.

## 3. Prognostic and Therapeutic Contributions of AML Genome Characterization

### 3.1. Molecular Subgroups of AML with Normal Karyotype

We have known for a long time that cytogenetic changes are among the most important prognostic factors in AML [45,46,47]. The previous prognostic classifications described three risk categories: favorable risk (recurrent translocations: t(15;17), t(8;21) or inv(16)), intermediate risk (normal karyotype) and unfavorable risk (complex karyotypes, monosomy 5, monosomy 7 or deletion 17p). Each category is associated with a different outcome, with 84% of complete remission (CR) in the favorable risk group, 76% in the intermediate and only 55% in the unfavorable risk groups [48].

Patients with normal karyotype represent 40–50% of all AML patients. Despite being grouped in the intermediate risk category, they present a heterogeneous prognostic. Molecular biology studies allowed a better prognosis stratification of these patients. Incidentally, Schlenk et al. showed that the mutational status of the *NPM1*, *FLT3*, *CEBPa*, *MLL*, and *NRAS* genes impacted the clinical outcome in a cohort of 872 adults younger than 60 years old with cytogenetically normal AML. Indeed, they highlighted that *NPM1* mutant patients without *FLT3* internal tandem duplication (FLT3-ITD), or *CEBPa* mutant patients, are associated with a decreased relapse risk (hazard ratio = 0.44 and 0.48, respectively). They also demonstrated that HSC transplantation (HSCT) is only beneficial for adverse genotypes (FLT3-ITD or wild-type *NPM1* and *CEBPa* without FLT3-ITD) [49]. Accordingly, ELN (European LeukemiaNet) recommended in 2010 rapid evaluation of the mutational status of *NPM1, FLT3* and *CEBPa* in normal karyotype AML patients to better characterize their prognosis risk and define the best therapeutic option [7].

With sequencing technologies progress, additional prognostically relevant recurrent somatic mutations have been identified. Patel et al. studied in 2012 the prognostic relevance of 18 genes in a large cohort of 398 AML patients. FLT3-ITD, *MLL* partial tandem duplication and mutations in *ASXL1* and *PHF6* were associated with a reduced overall survival in this study [50]. Gradually, with the identification of novel recurrent somatic mutations, novel molecular markers of outcome have been unraveled, allowing the definition of novel molecular prognostic classifications. In 2016, Papaemmanuil et al. found that the prognostic effects of individual mutations were impacted by the presence or absence of other somatic mutations. For example, co-occurrence of *ASXL1* and *SRSF2* mutations confers a particularly poor prognosis [32]. The impact of other driver mutations is well known for *NPM1* mutated AML. Indeed, *NPM1* mutations confer a favorable prognosis only in absence of a FLT3-ITD with a high allelic ratio [51,52].

Based on these molecular alterations recently associated with patient’s outcome, ELN defined in 2017 a new AML prognosis risk classification [7]. This latter defines treatment intensity, thus making HSCT only recommended for intermediate and unfavorable groups.

### 3.2. Minimal Residual Disease: An Independent Prognosis Factor

Cytogenetic and molecular characterization of AML at diagnosis allow a better risk stratification and treatment adaptation. In addition, measurement of minimal residual disease (MRD) after AML induction therapy has emerged as a major independent prognostic factor. First used for acute lymphoblastic leukemia response evaluation and risk stratification [53,54], it has more recently been developed in AML.

MRD corresponds to the persistence of a small number of residual leukemic cells, undetectable by morphological techniques, resistant to initial chemotherapy and often thought to be responsible for relapse. For AML, MRD can be assessed either by flow cytometry or by molecular biology techniques [55,56,57]. Molecular biology advances have improved MRD measurements with increasingly sensitive techniques, and have also broadened its application to a larger group of patients.

In AML, MRD was initially applied to the measurement of fusion transcripts: RUNX1-RUNX1T1, CBFb-MYH11 and PML-RARa; and then extended to the detection of residual *NPM1* mutated clones [58,59,60,61]. Fusion transcripts and *NPM1* mutations allow the measurement of MRD in 40 to 70% of AML patients. Real-time quantitative polymerase chain reaction (RT-qPCR) is currently the gold standard for such MRD quantification, with a sensitivity of 10^−4^ to 10^−6^. In the absence of a fusion transcript or a *NPM1* mutation, new approaches by digital PCR and next-generation sequencing (NGS) are being developed to evaluate the evolution of the various somatic mutations present at AML diagnosis [62,63]. However, not all mutations can be used as MRD markers. Signaling pathway mutations (*FLT3-ITD*, *TKD*, *N-RAS*, *KIT*) are unstable between diagnosis and relapse. Age-related mutations (*DNMT3A*, *ASXL1*) can persist at high rates in the remission state without any outcome predictive value [64]. Wilms’ tumor gene 1 (WT1) encodes a transcription factor overexpressed in 80% to 90% of AML cases [65]. High levels of WT1 after induction therapy have been associated with an increased relapse risk [66], while WT1 expression level before and after HSCT has been associated with post transplantation outcomes [67,68,69].

MRD is associated with response depth and is now used as an independent prognostic factor in adult AML clinical trials; in order to improve therapeutic stratification [70,71,72]. Accordingly, Balsat et al. showed the lack of benefit of HSCT in *NPM1* and FLT3-ITD mutated patients; who obtained a 4 log_10_ reduction of their *NPM1* MRD measured in remission blood samples [72]. MRD determined by molecular biology techniques has also been associated with patients’ relapse [73,74,75]. Detection of RUNX1-RUNX1T1 transcript in the blood of CBF AML patients with t(8;21) in remission is indeed associated with a four-years relapse risk of 87% [76].

### 3.3. New Molecular Anomalies as Novel Therapeutic Targets

For a long time, AML induction treatment was exclusively based either on intensive chemotherapy combining an anthracyline with cytarabine (“3+7” regimen) for young fit patients, or on low intensity therapeutic regimens such as low dose cytarabine or hypomethylating agents (HMAs) for unfit or elderly patients. If this first treatment resulted in a complete remission, consolidation treatment with cytarabine was initiated and was followed by HSCT for fit patients with the more aggressive AML subtypes [8].

More recently, AML treatment entered a new era thanks to a better physiopathological understanding. Advances in molecular biology technologies allowed identification of new driver mutations. These findings not only improved prognostic stratification but also opened the road for the development of novel targeted therapies in AML.

The use of these therapies in AML is not a new story. Indeed, it dates back to the use of all-transretinoic acid (ATRA) and arsenic trioxide for Acute Promyelocytic Leukaemia (APL) treatment in the 1980s. Collaborative efforts between Chinese and French teams demonstrated that ATRA and arsenic were able to induce complete remissions in APL patients [77,78]. These two therapies were initially tested empirically, without knowing their underlying antileukemogenic mechanism. In 1990, De Thé et al. finally cloned the breakpoint of the t(15;17) translocation, pathognomonic of APL [79]. They then discovered the PML-RARa fusion gene, which encodes the fusion protein responsible for leukemogenesis in APL. This oncoprotein interferes with the two normal proteins and is thought to be responsible, through its RARa domain, for a dominant deregulation of the normal effect of the wild-type RARa protein on cellular differentiation. We thus observe a blockage in promyelocytes maturation and an increase in their proliferation and self-renewal. The PML domain of the PML-RARa fusion protein prevents the assembly of nuclear bodies. In the presence of the PML-RARa fusion protein, DNA transcription is impaired through the binding of nuclear corepressors to RARa. Therefore, the PML-RARa fusion protein initiates leukemogenesis by disabling proliferating blasts from differentiating. ATRA and arsenic work synergistically block this oncogenic process. ATRA prevents this inhibition by binding to the RARa domain. Importantly, both ATRA and arsenic trioxide induce proteasome-mediated degradation of the PML-RARa fusion protein, therefore representing the first example of a synergistic targeted therapy in cancer [80,81,82,83,84].

Symmetrically to the APL success story, the identification of new mutations in genes such as *FLT3* or *IDH1/2,* and the understanding of their leukemogenic impact, enabled the development of new targeted therapies. As opposed to the ATRA/arsenic combination, *FLT3* and *IDH1/2* inhibitors were developed after the pathophysiologic mechanism underlying the role of these mutations in AML was understood.

*FLT3* mutations in AML were first described in the 2000s. Two types are pathogenic in AML: FLT3-ITD (Internal Tandem Duplication) and FLT3-TKD (Tyrosine kinase Domain) mutations, for which the prevalence is respectively 20–25% and 7–9% in adult patients. FLT3-ITD mutations correspond to the insertion of 3 to 400 base pairs either within exon 14 (70%) encoding the juxta-membrane domain of the FLT3 receptor, or within exon 15 (30%) encoding the first tyrosine kinase domain. FLT3-TKD-like mutations are point mutations that occur within the second tyrosine kinase domain [85]. These two types of mutations constitutively activate the FLT3 receptor; leading to activation of the intracellular PI3K/AKT, MAPK and STAT5 downstream pathways. This ultimately results in increased cellular proliferation and resistance to apoptosis [86]. Stirewalt et al. showed in 2002 that leukemic cells with *FLT3* mutations are able to lose their wild-type allele through loss of heterozygosity, arguing for an oncogenic addiction [87]. Additionally, AML cells with *FLT3* mutations are thought to be resistant to conventional chemotherapy and are found in 75% of cases after relapse, therefore making the mutated FLT3 receptor a relevant therapeutic target to improve long-term remission in AML.

Thus, various inhibitors of the mutated FLT3 receptor have been developed. They belong to three generational and two functional categories (type I inhibitors which inhibit the FLT3 receptor in its two conformations: active and inactive kinase, and type II inhibitors impairing the FLT3 receptor only in its inactive kinase conformation). Midostaurine, a type I multi-kinase inhibitor was the first inhibitor approved by the Food and Drugs Administration (FDA) in AML. In 2005, the RATIFY trial proved its efficacy in combination with first-line chemotherapy, with a four-year overall survival of 51.4% in the interventional arm and 44.3% in the standard chemotherapy arm (*p* = 0.0078) [88]. Since then, many trials have demonstrated the efficacy of Gilteritinib, a third generation FLT3 inhibitor recently approved in refractory or relapsed FLT3 mutated AML [89,90,91,92,93]. Further clinical trials are ongoing to evaluate the impact of new generation FLT3 inhibitors in combination with first-line chemotherapy.

Mutations of isocitrate dehydrogenase 1 and 2 (*IDH1* and *IDH2*) were described for the first time in AML in 2009. *IDH1* mutations affect 7 to 15% of AML patients, while *IDH2* mutations are present in 7 to 20% patients [94]. They correspond to nonsense mutations involving the enzyme’s active site in arginine-rich regions at codon 132 for *IDH1* and codons 140 or 172 for *IDH2*. The IDH1 and two mutants lose their normal reduction activity and oxidative function and preferentially produce an oncometabolite named 2-hydroxyglutarate (2-HG). This oncometabolite competes with enzymes regulating cellular methylation, resulting in hypermethylation of DNA and histones and impaired cellular differentiation. Therefore, *IDH1* and *2* mutations play a key role in leukemogenesis and represent ideal therapeutic targets. IDH1 and IDH2 inhibitors lead to a significant reduction in the production of 2-HG and ultimately allow the resumption of cellular differentiation. In refractory or relapsed AML, IDH1 (Ivosidenib) and IDH2 (Enasidenib) inhibitors have shown interesting results in monotherapy with response rates of 31% and 40%, respectively [95,96].

Mutations of the tumor protein p53 (*TP53*) gene were first described in AML patients by Fenaux et al. in 1991 [97]. They are present in 5–10% of AML and are mostly found in elderly patients in complex karyotype and therapy-related AML [98,99,100,101]. *TP53* mutations confer resistance to standard chemotherapy, and the prognosis of *TP53*-mutated AML is very poor [32,102]. HMAs are used as first-line treatment for *TP53*-mutated AML with a median overall survival (OS) of 7.2 months [103]. Discovery of novel therapies for *TP53*-mutated AML, therefore, remains an unmet clinical need [104]. APR-246, a small molecule which restores the wild-type TP53 conformation and activity in *TP53*-mutated cells, has shown interesting results in a phase Ib/II clinical trial [105,106]. In this trial, APR 246 associated with AZA in 11 adult patients with *TP53* mutant oligoblastic AML (<30% blasts), showed an ORR of 64% [107]. A phase III clinical trial is currently ongoing to further evaluate the effects of APR-246 in *TP53* mutated AML (NCT03745716). However, targeting these single mutations may result in resistance mechanisms. Further clinical trials are ongoing to validate these therapies efficacy in combination with standard of care AML therapeutics (chemotherapy, hypomethylating agents).

Together with targeted therapies, two novel treatments have recently emerged, resulting in improved AML therapeutic strategies for specific patient subgroups. Vyxeos, a liposomal combination of daunorubicin and cytarabine was approved in 2017 for patients with therapy-related AML or AML with myelodysplastic-related-changes [108]. Venetoclax, a small oral inhibitor of the antiapoptotic protein BCL2, has significantly widened the therapeutic landscape for older unfit patients. Indeed, a randomized-phase III trial in untreated AML patients ineligible for standard therapy (unfit or older than 75 years), showed an OS improvement with the association of Venetoclax and 5-azacytidine compared to placebo and 5-azacytidine (HR = 0.66%; *p* < 0.001) [109].

Table 1 highlights the clinical relevance of the main contributions of descriptive genomics in AML.

This table summarizes genetic alterations in AML that have clinical relevance nowadays. They are commonly used to predict prognosis and decide for therapeutic intensity. Some of them are used for following MRD or/and represent therapeutic targets.

## 4. Functional Genomics: A Better Understanding of the Functionality and Interactions between Different Oncogenes in Aml, towards New Therapeutic Avenues

### 4.1. Therapeutic Limitations of Descriptive Genomics in AML

Descriptive genetic characterization of AML has several therapeutic limitations.

Many frequently mutated genes are essential genes that cannot be targeted without intolerable cellular damage. For example, the *NPM1* gene encodes a chaperone protein essential for ribosomal synthesis in nonleukemic cells. The therapeutic window for *NPM1* targeting might thus be very narrow.

Other commonly mutated genes in AML such as transcription factor are not pharmacologically druggable and represent a chemical challenge nowadays.

Additionally, due to AML’s complex subclonal heterogeneity, targeting a single mutated gene leads to the development of resistance mechanisms. Indeed, leukemic cells have an impressive adaptive capacity enabling them to develop diverse strategies to escape the therapeutic pressure imposed by targeted therapies. For example, Quek et al. showed that relapse after IDH2 inhibition results from clonal evolution or selection of terminal or ancestral clones. The reoccurrence of differentiation arrest can be due to seven different mutational mechanisms [110]. Moreover, Intlekofer et al., demonstrated that after exposure to the IDH2 inhibitor Enasidenib, leukemic cells were able to acquire a new mutation on the wild type *IDH2* allele responsible for steric hindrance at the Enasidenib binding site, and ultimately for resistance to IDH2 inhibition [111].

Due to these frequent resistance mechanisms, it is now admitted that targeted therapies should be included in the AML treatment strategy as part of wisely designed therapeutic combinations. Several clinical trials have tested the combination of IDH2 inhibitors with chemotherapy or hypomethylating agents. This induces better response rates than monotherapy. In fact, the combination of Ivosidenib or Enasidenib with standard induction chemotherapy achieves higher complete remission (CR) rates compared to monotherapy alone (86% and 40%, respectively). The association of Enasidenib with 5-azacytidine in first line IDH-mutated AML also shows interesting results with 71% overall response rate (ORR) for the combination, compared to 42% for 5-azacytidine alone (*p* = 0.0064) [112]. The combination of three molecules Venetoclax, Azacytidine and Ivosidenib, was recently presented by DiNardo and colleagues at the European Hematology Association conference. This frontline association seems very promising with 100% ORR, 75% CRs and 67% of patients with negative MRD [113].

Another resistance mechanism could stem from the hierarchical organization of AML cells in different subpopulations, and the presence of more immature cells called Leukemic Stem Cells (LSCs) known to maintain leukemogenesis and potentially lead to patient relapse [114,115]. Hence, this subpopulation represents a promising target for developing new therapeutic strategies to obtain a definitive AML cure. Nevertheless, LSCs, defined by their ability to initiate leukemia in immune-compromised murine models [116], may express different surface markers across patients and sometimes even within the same patient, making their isolation and identification more challenging. Many of the genetic alterations reported in this review are associated with differential expression of cell surface markers in leukemic cells. FLT3 mutations have, for example, been associated with the presence of a CD34+CD123+CD25+CD99+ immnuophenotype [117]. DNMT3A, NPM1, and FLT3-ITD triple-mutated AML cells are characterized by presence of a CD34^low^ GPR56^high^ immunophenotype [118]. Additionally, it has been recently shown that LSCs suppress NKG2DL surface proteins necessary for NK mediated killing of leukemic cells, subsequently enabling LSCs immune escape [119]. These examples offer evidence that together with genetic alterations, leukemia-associated immunophenotypes (LAIP) represent a valuable tool in the clinical setting, either to assess relapse risk in AML patients or to decrease this risk through specific therapeutic targeting. Indeed, the gemtuzumab ozogamicin antibody drug conjugate is an example of antigen targeted therapy directed against the CD33 surface marker. However, healthy hematopoietic cells also express this marker, therefore decreasing this strategy’s specificity [120,121,122,123,124]. Ongoing clinical trials are investigating other cell surface targets such as CD47, CD123 and CLL-1, which may improve LSCs targeting; leading to better clinical outcomes in AML patients [125,126,127].

Finally, leukemogenesis does not only depend on the intrinsic accumulation of genetic alterations responsible for the progression of normal HSCs into leukemic stem cells, but it also relies on an intensive crosstalk between AML cells and their osteomedullary niche. Indeed, some patients develop a new leukemia derived from the donors’ stem and progenitor cells, several years after HSC transplantation for an initial AML [128]. This clinical observation illustrates the key role of the tumor microenvironment in leukemic transformation. In addition to the influence of the bone marrow niche on haematopoiesis, leukemic cells have been involved in microenvironment remodeling in order to promote their own growth through a dynamic bilateral relationship between the “seeds” and the “soil” [129]. Adhesion proteins expression variations, clonal chromosomal abnormalities, hypoxia or vascular permeability play a key role in the initiation, development and chemoresistance of AML [130,131,132,133,134]. Descriptive genetics can, therefore, not fully recapitulate the complex leukemogenic process. Functional genetics studies should help dissect this niche-leukemic crosstalk and identify new leukemic cell dependencies on their microenvironment. Concomitant targeting of the “seeds” and the “soil” could ultimately help eradicating leukemic cells and improve patients’ survival.

### 4.2. New Models and New Technologies to Better Understand AML Leukemogenesis and Ultimately Improve Treatment

Advances in genetic technologies have allowed the identification of new anomalies and an increasingly precise molecular characterization of AML. To better understand the functional impact of these genetic abnormalities in leukemogenesis, cellular and animal models, as well as genetic engineering tools, have been gradually developed over the past decades. From the purely descriptive genetics time, we have now entered an exciting functional genetics era.

Functional genetic studies were initially developed to understand the leukemogenic mechanisms induced by genomic alterations using in vitro cellular models of AML. These cellular models were derived from patient leukemic cells after immortalization. This first step towards the discovery and validation of the oncogenic pathophysiological hypothesis remains a simplistic model that does not recapitulate the AML development complexity within a remodeled bone marrow niche.

Scientists then turned to nonmammalian animal models to study the mechanisms of leukemogenesis in a more complex setting in vivo. For example, drosophila was initially used to study the molecular and phenotypic consequences of the translocation t(8;21) in leukemic cells. The simplicity of drosophila’s handling, and its similarities with mammals in terms of haematopoiesis, made it an ideal model [135,136]. Using RNA interference screening approaches, Osman et al. were able to show the key role of the CalpainB protein in the leukemogenesis induced by the fusion protein AML1-ETO. In drosophila, inhibition of CalpainB led to both the elimination of the AML1-ETO fusion protein and the absence of clonal proliferation [137]. The mechanistic underpinnings of the translocation t(8;21) have also been studied in the zebrafish model, which is easy to handle and shares many genetic and molecular mechanisms with the human species [138]. This model allowed wide screening of small inhibitory molecules. Cox2 signaling inhibition or b-catenin pathway impairment were, for example, shown to antagonize the effects of the AML1-ETO fusion protein on hematopoietic differentiation. This observation could potentially have an implication in patients with this AML subtype [139].

Although having contributed to many discoveries, the main limitation of these animal models remains their microenvironment, which is very different from that of the human species. Transposing the results obtained using such models to humans is, therefore, challenging.

To overcome this lack of resemblance, studies of AML pathophysiology turned to mouse models. These are not only manageable research models but also present genetic and physiological similarities with the human species, making them an acceptable model to approach human disease mechanisms.

Syngeneic mouse models were thus developed to study the evolution of murine tumor cells within their own murine microenvironment. Murine tumorigenesis can be obtained through different mechanisms: chemoinduced, radioinduced, viroinduced or using transgenic tools. These syngeneic models have, for example, participated in the understanding of chemotherapeutic treatment mechanisms [135,140,141] or ATRA/arsenic mechanistic effects in APL models [142,143]. They offer the advantage of studying the effect of the tumor microenvironment on leukemic growth. However, the use of murine tumor cells that are biologically very different from human ones does not guarantee a good correlation between the therapeutic activity of some molecules in syngeneic models and their efficacy in humans.

To avoid this lack of correlation related to interspecies differences, scientists developed xenograft models. These enable the study of leukemogenesis using human cancer cells in immune-deficient mice. Different immune-deficient mouse strains have been engineered over the past years: SCID mice (Severe Combined Immune Deficient mice, lacking B and T cell immunity due to an autosomal recessive mutation in the *PRKDC* gene), NOD/SCID mice (Nonobese Diabetic SCID mice characterized by an impairment of NK activity, reduced mature macrophages and total lack of B and T cells) followed by NSG mice more recently (NOD/SCID gamma mice generated through deletion or truncation of the IL-2R gamma chain, harboring a complete abolition of innate and adaptive immunity). Gradually, mechanisms of immune-suppression evolved in order to allow better human leukemic engraftment. These models were at the origin of the discovery of the concept of leukemia-initiating cells (LIC) and their progeny [135,144,145]. Although crucial for human AML cell engraftment, the main limitation of these models remains precisely their immune-suppression. The immunosuppression induced to permit engraftment incidentally abrogates potential microenvironment effects on leukemogenesis. Moreover, interspecies differences may impair leukemic-niche crosstalk mechanisms between the human AML cells and the murine bone marrow niche.

Very recently, humanized models have been developed to overcome the latest limitation. In these sophisticated models, human leukemic cells evolve within a humanized bone marrow microenvironment. These humanized leukemic organoids implanted in immune-deficient mice are, therefore, susceptible to reproduce with the greatest fidelity the mechanisms of leukemogenesis observed in the human species. These models indeed allow a better engraftment of primary AML cells while maintaining the clonal diversity observed in the original patients [146,147].

The main advantages and pitfalls of the evolving AML models are summarized in Figure 4.

In parallel to AML modelling advances, new molecular biology and bioinformatics tools have been developed to better understand the functionality of the different oncogenes involved in AML and discover novel therapeutic opportunities.

The short hairpin RNA (shRNA) techniques established in the 2000s allowed the extinction of target genes. Cellular expression of shRNAs can be achieved through plasmid expressing viral vectors. Once the vector is integrated into the host genome, the shRNA is transcribed and exported to the nucleus. After its loading into the RISC (RNA-induced silencing complex) complex, the sense strand is degraded and the antisense strand complexes with the complementary mRNA’s sequence. The target gene is silenced either by mRNA cleavage or translation failure. ShRNA approaches allowed many advances in the understanding of AML leukemogenesis and its therapeutic implications. For example, Zuber et al., Fenouille et al. and Benajiba et al. were able to demonstrate, using large-scale shRNA functional screening approaches, that inhibition of the bromodomain protein Brd4 leads to AML cell growth impairment on one hand, and that inhibiting the mitochondrial creatine kinase (CKMT1) pathway on the other hand leads to proliferation arrest and apoptosis in the EVI1-driven AML subtype [148,149,150]. Targeting the bromodomain family (BET) and CKMT1 are thus promising therapeutic strategies for AML.

The recent discovery of the CRISPR-Cas9 technology allowed further understanding of AML physiopathology and novel therapeutic considerations. The CRISPR-Cas9 system, which includes a guide RNA coupled to an endonuclease, is originally used by bacteria for antiviral natural defense. In 2010, Charpentier et al. adapted this tool for inactivation of target genes or replacement of mutated genes with a healthy copy. In the cancer field, this technology opened the road for functional understanding of various genetic alterations and identification of novel cancer dependencies. In AML for instance, Tzelepis and colleagues unraveled that impairment of the *KAT2A* gene was responsible for leukemic growth inhibition and, therefore, represents a potential therapeutic target [151]. Similarly, Yamauchi et al. used a genome-wide CRISPR-Cas9 screening approach to show that the mRNA decapping enzyme scavenger DCPS is essential for AML cells survival [152].

Finally, to better understand the functional impact of the multiple mutations involved in carcinogenesis, several genome-wide large scale comprehensive studies have been recently initiated. The cancer dependency map (DEPMAP Project), a large-scale genetic integrative project developed by the Broad Institute is an example of such community efforts [1,153]. It is now accepted that some acquired mutations in genes, such as *FLT3* or *IDH1/2*, represent therapeutic targets. The main therapeutic challenge resides in the fact that most of the genomic alterations described in this review remain nondruggable. Identifying the cellular dependencies induced by these genomic alterations may offer the key to answer this challenging question. The aim of the DEPMAP project, using hundreds of cancer cell lines and whole-genome approaches, is to precisely create a comprehensive map of cancer dependencies. Such strategies will indeed help defining a novel landscape of therapeutic targets, identifying patients who could respond to these therapies, and developing a better understanding of leukemogenic mechanisms and AML vulnerabilities.

## 5. Conclusions

Over the past decades, descriptive and functional genomics have substantially improved the understanding of AML leukemogenesis. From the definition of molecular prognosis classifications to the development of novel targeted therapies, these advances have contributed to shaping a novel and more efficient therapeutic landscape in AML.

Further studies allowed by the recent technological and modelling advances described in this review will be key for a more comprehensive understanding of AML initiation, proliferation and therapy resistance processes. Wisely combining functional and descriptive genomic tools with physiopathologically relevant disease modeling, holds the promise for the collaborative construction of a solid road towards an AML cure within the next decades.

## Figures and Tables

**Figure 1 cancers-13-00748-f001:**
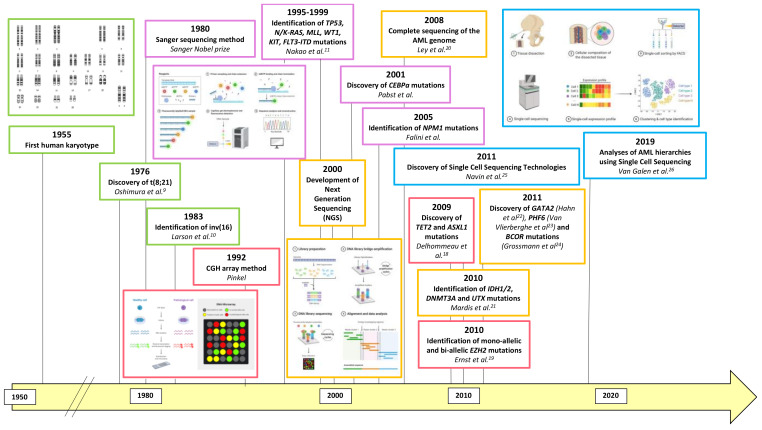
The history of AML genomic discoveries. Green, purple, red, orange and blue represent respectively cytogenetic, Sanger sequencing, CGH array, NGS and Single Cell Sequencing technological discoveries. CGH: Comparative Genomic Hybridization; NGS: Next Generation Sequencing; AML: Acute Myeloid Leukemia.

**Figure 2 cancers-13-00748-f002:**
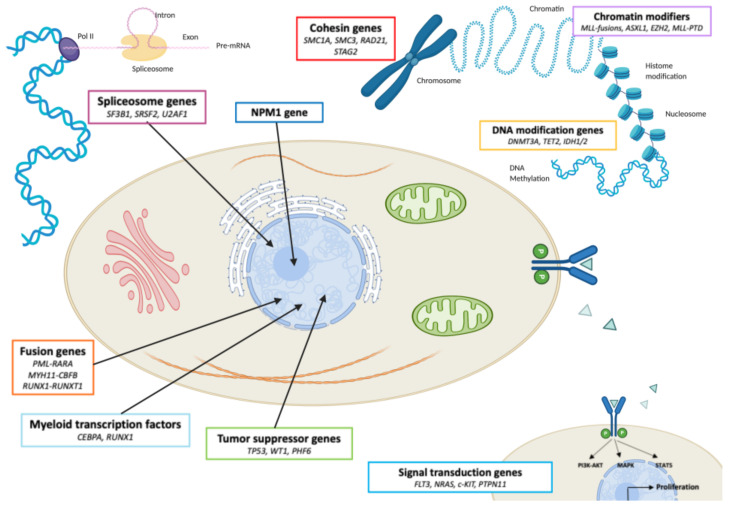
Functional categories of AML mutated genes. Genes mutated in AML are grouped based on their biological function. Gene categories are schematically depicted at the cellular level.

**Figure 3 cancers-13-00748-f003:**
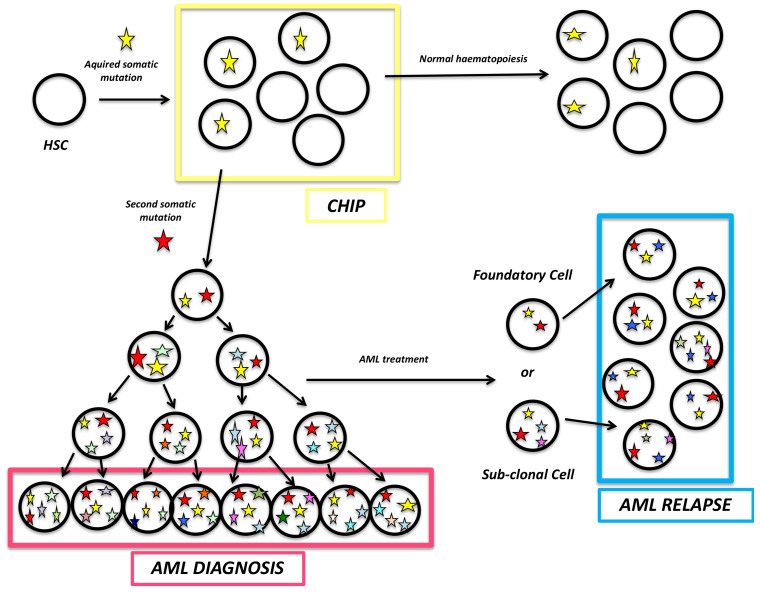
Leukemogenesis and AML clonal evolution: from preleukemic state to relapse. HSC: Hematopoietic Stem Cell; CHIP: Clonal Haematopoiesis of Indeterminate Potential; AML: Acute Myeloid Leukaemia.

**Figure 4 cancers-13-00748-f004:**
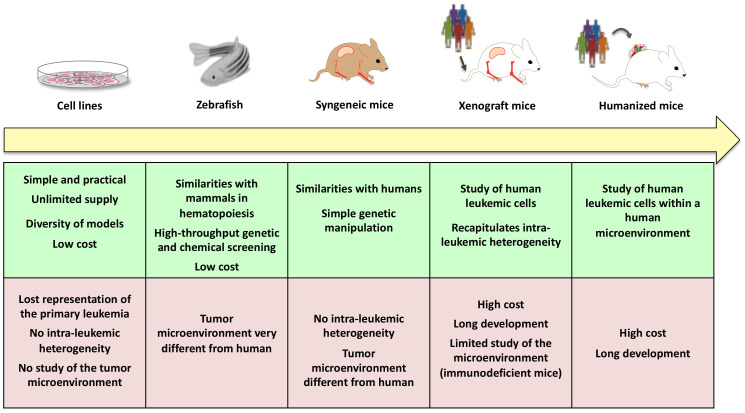
AML models over time: main advantages and drawbacks. This schema highlights several models used throughout the years (yellow arrow) for a better understanding of AML leukemogenesis. Green and purple boxes represent respectively the main advantages and drawbacks of each AML model.

**Table 1 cancers-13-00748-t001:** Clinical relevance of descriptive genomics in AML. Examples of genetic alterations, their prognostic impact and their potential use as a response MRD marker and/or therapeutic target.

Molecular Marker	Prognostic Impact	Marker of MRD	Targeted Therapy
PML-RARa	FAVORABLE	Yes	ATRA and Arsenic Trioxide
RUNX1-RUNX1T1	FAVORABLE	Yes	None
CBFb-MYH11	FAVORABLE	Yes	None
NPM1 mutant	without FLT3-ITD or with FLT3-ITD low ratio FAVORABLE	Yes	None
	with FLT3-ITD high ratio INTERMEDIATE		
FLT3 mutant (ITD or TKD)	FLT3-ITD low ratio with NPM1 wild type FLT3-ITD high ratio with NPM1 mutant INTERMEDIATE	No	FLT3 inhibitors (1) Three generational categoriesMIDOSTAURINE (1st) GILTERITINIB (3rd) (2) Two functional categories
	FLT3-ITD high ratio with NPM1 wild type UNFAVORABLE		
IDH1/IDH2 mutant	None	No	IVOSIDENIB (IDH1 inhibitor) ENASIDENIB (IDH2 inhibitor)

ITD: Internal Tandem Duplication; TKD: Tyrosine Kinase Domain; MRD: Minimal Residual Disease.

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
