# Peer review of "Descriptive and Functional Genomics in Acute Myeloid Leukemia (AML): Paving the Road for a Cure"

_cancers, 2021, doi:10.3390/cancers13040748_

Round 1

Reviewer 1 Report

The review written by Hélène Pasquer et al is suitable for publication being a comprehensive paper. However, I had some recommendations that may increase the interest of the readers.

The authors may describe in details the therapies for AML and the differences between elderly and young AML patients. Also, even if is not the aim of the paper, the importance of marker expression compared to genetics may be shortly noticed.

The papers published by Cancers Journal - Recently Approved Therapies and Drugs in Development by Michele Stanchina , Acute Myeloid Leukemia: Aging and Epigenetics by Polina Zjablovskaja et al and Acute Myeloid Leukemia Stem Cells: The Challenges of Phenotypic Heterogeneity by Marlon Arnone may be useful (for example).

One Graphical abstract may increased the interest of the readers.

Reviewer 2 Report

The manuscript is well written, it is readable. The authors describe a step-by-step genetic investigation from history to state-of-the-art methods that have enabled a better understanding of AML leukemogenesis. This led to direct clinical implications, namely determination of prognosis, MRD monitoring and the introduction of new targeted therapies. The paper contains brief / clear summaries in the form of schematic diagrams or tables.

I only recommend changing the information on the prognostic importance of the expression status of PB mRNA WT1 and supplementing the data according to the latest studies, e.g. Salek C. Clin Lymphoma Myeloma Leuk. 2020; Goel H. Am J Blood Res. 2020;  Rautenberg C. Blood Cancer J. 2019.
